# *PISD* is a mitochondrial disease gene causing skeletal dysplasia, cataracts, and white matter changes

Tian Zhao[1,2,*], Caitlin M Goedhart[1,*], Pingdewinde N Sam[3], Rasha Sabouny[1,2] , Susanne Lingrell[4], Adam J Cornish[3], Ryan E Lamont[1], Francois P Bernier[1,6], David Sinasac[1], Jillian S Parboosingh[1], Care4Rare Canada Consortium, Jean E Vance[4], Steven M Claypool[3] , A Micheil Innes[1,5,6] , Timothy E Shutt[1,2,5]

Exome sequencing of two sisters with congenital cataracts, short stature, and white matter changes identified compound heterozygous variants in the *PISD* gene, encoding the phosphatidylserine decarboxylase enzyme that converts phosphatidylserine to phosphatidylethanolamine (PE) in the inner mitochondrial membrane (IMM). Decreased conversion of phosphatidylserine to PE in patient fibroblasts is consistent with impaired phosphatidylserine decarboxylase (PISD) enzyme activity. Meanwhile, as evidence for mitochondrial dysfunction, patient fibroblasts exhibited more fragmented mitochondrial networks, enlarged lysosomes, decreased maximal oxygen consumption rates, and increased sensitivity to 2-deoxyglucose. Moreover, treatment with lyso-PE, which can replenish the mitochondrial pool of PE, and genetic complementation restored mitochondrial and lysosome morphology in patient fibroblasts. Functional characterization of the *PISD* variants demonstrates that the maternal variant causes an alternative splice product. Meanwhile, the paternal variant impairs autocatalytic self-processing of the PISD protein required for its activity. Finally, evidence for impaired activity of mitochondrial IMM proteases suggests an explanation as to why the phenotypes of these *PISD* patients resemble recently described "mitochondrial chaperonopathies." Collectively, these findings demonstrate that *PISD* is a novel mitochondrial disease gene.

## Introduction

Mitochondria are double-membrane–bound organelles, which in addition to generating most of a cell's energy via oxidative phosphorylation, have important roles in regulating many other cellular

processes (e.g., apoptosis, immune response, and numerous metabolic pathways [Nunnari & Suomalainen, 2012]). Although mitochondrial dysfunction has been implicated in a growing list of human diseases, more severe forms of mitochondrial dysfunction cause a group of rare disorders known as mitochondrial diseases, estimated at ~1 in 4,300 in adults (Gorman et al, 2015). Classic mitochondrial disease is caused by impaired energy production, and often manifests in tissues with high energy demands, such as heart, muscle, brain, and eyes. However, diagnosing mitochondrial disease is difficult because of the clinical and genetic heterogeneity of this group of disorders.

More recently, an atypical class of mitochondrial diseases has been described where impaired mitochondrial protein homeostasis appears to be the underlying cause of mitochondrial dysfunction (Royer-Bertrand et al, 2015). These "mitochondrial chaperonopathies" are characterized by atypical skeletal phenotypes and craniofacial features that are not commonly seen in classic mitochondrial disease, as well as cataracts and central nervous system involvement, which are sometimes found in mitochondrial disease. To date, only three genes (*LONP1*, *HSPA9*, and *AIFM1*) encoding mitochondrial proteins have been linked to mitochondrial chaperonopathies (Dikoglu et al, 2015; Royer-Bertrand et al, 2015; Strauss et al, 2015; Mierzewska et al, 2016). The molecular mechanisms through which impaired mitochondrial protein homeostasis lead to skeletal abnormalities, rather than more traditional mitochondrial disease phenotypes, remains unknown.

Maintenance of mitochondrial protein homeostasis is, thus, a key aspect regulating mitochondrial function, and its impairment leads to disease. Notably, many mitochondrial-specific proteases are bound to the inner mitochondrial membrane (IMM) (Quiros et al, 2015). Thus, it is not surprising that the IMM lipid composition is an important regulator of mitochondrial function (Lu & Claypool, 2015). The *PISD* gene encodes a mitochondrial-localized enzyme that converts phosphatidylserine (PS) to phosphatidylethanolamine

[1]Alberta Children's Hospital Research Institute, Department of Medical Genetics, Cumming School of Medicine, University of Calgary, Calgary, Alberta, Canada [2]Department of Biochemistry and Molecular Biology, Cumming School of Medicine, University of Calgary, Calgary, Alberta, Canada [3]Department of Physiology, Johns Hopkins University School of Medicine, Baltimore, MD, USA [4]Department of Medicine and Group on Molecular and Cell Biology of Lipids, University of Alberta, Edmonton, Alberta, Canada [5]Hotchkiss Brain Institute, Cumming School of Medicine, University of Calgary, Calgary, Alberta, Canada [6]Department of Pediatrics, Cumming School of Medicine, University of Calgary, Calgary, Alberta, Canada

Correspondence: micheil.innes@ahs.ca;; timothy.shutt@ucalgary.ca
*Tian Zhao and Caitlin M Goedhart contributed equally to this work.

(PE) in the IMM (Percy et al, 1983; Zborowski et al, 1983; Calvo et al, 2016; Smith & Robinson, 2018). PE, which comprises ~15–25% of cellular membranes, is an important lipid that provides membrane curvature (Vance & Tasseva, 2013). Although complete loss of *PISD* is embryonic lethal in mice, highlighting the importance of mitochondrial PE, heterozygous mice do not have any overt phenotypes (Steenbergen et al, 2005). In cellular models, severe depletion or complete loss of phosphatidylserine decarboxylase (PISD) results in decreased mitochondrial oxidative phosphorylation and fragmentation of the mitochondrial network (Steenbergen et al, 2005; Tasseva et al, 2013). Notably, an autocatalytic processing event that generates two subunits ($α$ and $β$) is required to form a functional PISD enzyme (Li & Dowhan, 1988).

In the present study, we report the first example of patients with pathogenic variants in *PISD*, who presented with congenital cataracts, short stature, mid-face hypoplasia, hypomyelination, ataxia, and intellectual disability. These phenotypes are reminiscent of the skeletal abnormalities described for pathogenic variants in *LONP1* (cerebral, ocular, dental, auricular, skeletal syndrome [CODAS] syndrome), *HSPA9* (epiphyseal, vertebral, ear, nose, plus associated findings [EVEN-PLUS] syndrome), and *AIFM1* (spondyloepimetaphyseal dysplasia with mental retardation [SEMD-MR]), rather than classic mitochondrial disease. Our findings show that mitochondrial protein homeostasis is impaired in fibroblasts from patients with PISD variants. As such, we suggest that *PISD* be included in the list genes associated with impaired mitochondrial protein homeostasis.

# Results

## Clinical data

### Affected individual 1 (II-1) (Fig 1A)

This individual is now 28-y old. She was born at 36 wk gestational age after an uneventful pregnancy. Congenital cataracts were diagnosed in the first few months of life and were extracted surgically. At that time, the child was generally well, with a normal height and a basic metabolic workup (including screening for galactosemia), which was essentially normal. However, falloff in growth percentiles began in the first year of life. When reassessed at 3 y of age, there was evidence of mild developmental delay and recurrent respiratory infections, and height was now at −4 to −5 SDs below the mean. A skeletal survey carried out at 8 y of age was nondiagnostic with findings including brachydactyly, delayed bone age, mild thoracic platyspondyly, and metaphyseal striations not suggestive of a primary skeletal dysplasia but suggestive of bone hypoproliferation in keeping with an underlying genetic or metabolic syndrome. Increasing difficulty in school became evident, and subsequent psychometric testing was consistent with either mild intellectual disability or borderline intelligence. Serial Magnetic Resonance Imaging (MRI) scans revealed diffuse T2 signal intensity throughout the bihemispheric white matter, with progressive volume loss and hypomyelination of the corpus callosum (Fig 1B).

She was diagnosed with tracheal stenosis and required a temporary tracheostomy from age 10 to 11 in the context of an acute deterioration with an infectious illness. She was diagnosed at age 18 with bilateral progressive hearing loss. Her final adult height is 124.7 cm (−5.3 SD). She

had a distinctive facial appearance with depressed nasal ridge and midface hypoplasia (Fig 1A). She has been generally stable in adulthood. She briefly attended community college. She has tracheal stenosis and stridor but tends to be generally active and well. Her hearing loss has been slowly progressive.

### Affected individual 2 (II-2)

This is the younger sister of II-1, and these are the only two children born to healthy non-consanguineous parents. She was also diagnosed with congenital cataracts in early infancy. Her early trajectory followed a remarkably similar course to that of her elder sister. She also had multiple respiratory illnesses, some requiring ICU admission, mild global developmental delay, and progressive short stature. In addition, she developed an intention tremor in early childhood. She has a history of anxiety. She has subglottic stenosis and tracheomalacia and had an ICU admission for a respiratory decompensation at age 25. She was also recently diagnosed at age 25 with mild bilateral sensorineural hearing loss. Skeletal survey and MRI findings were similar to those in her elder sister. Her final adult height is 123.7 cm (−5.4 SD). She had midface hypoplasia and a depressed nasal ridge, similar to her elder sister (Fig 1A).

The presence of a remarkably similar and striking phenotype in two sisters born to unaffected parents suggested a likely autosomal recessive disorder. The findings, including white matter changes on MRI scan (Fig 1B), congenital cataracts, and progressive hearing loss, were potentially suggestive of a progressive neurogenetic or neurometabolic condition. However, as with her elder sibling, metabolic investigations were normal. Metabolic tests carried out in one or both have included urine metabolic screen, erythrocyte Gal-1-P uridyl transferase activity, galactokinase activity, and plasma amino acids. Clinical diagnoses on the differential diagnosis that all appeared unlikely included Sjogren–Larsson syndrome, Conradi–Hunnermann syndrome, hypomyelination–cataract syndrome, and Cockayne syndrome. The possibility, albeit unlikely, that these two siblings both shared more than one rare genetic condition was also considered; however, it was deemed most likely that they had a novel, previously undescribed genetic condition.

## Variant analysis

Whole-exome sequencing was performed on DNA extracted from peripheral blood provided by both sisters and their mother. Assuming an autosomal recessive inheritance pattern, we searched for rare coding homozygous or compound heterozygous variants (ExAC minor allele frequency less than 1% and observed in five or fewer other Care4Rare exome projects) that were shared by the sisters. The sisters did not share any rare homozygous variants, but they did share two different rare heterozygous variants in the genes *PISD*, *actin-related protein T1 (ACTRT1)* and *SON DNA-binding protein (SON)*. However, the mother carried a single variant, NM_001326411.1 (*PISD*):c.697+5G>A [p.(?)], in only one of the three genes. This confirmed that the sisters' shared variants in *PISD*, NM_001326411.1(*PISD*): c.830G>A [p.Arg277Gln] and NM_001326411.1(*PISD*):c.697+5G>A [p.(?)], were bi-allelic (Fig 1C) and that the remaining variants in ACTRT1 and SON were most likely inherited in cis.

The missense variant, c.830G>A [p.Arg277Gln], altered a highly conserved nucleotide (phyloP: 5.69 [−14.1; 6.4]) and amino acid

**A**

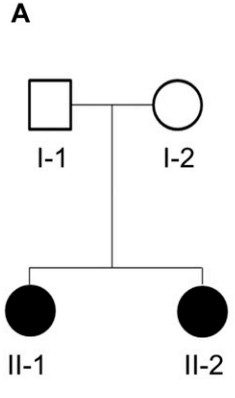

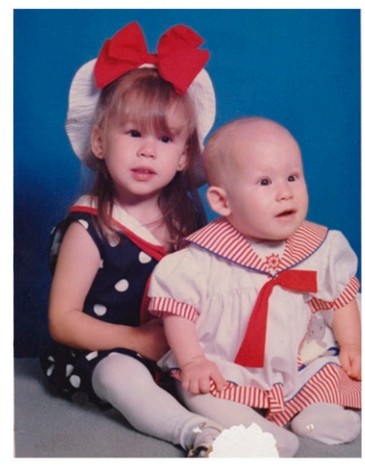

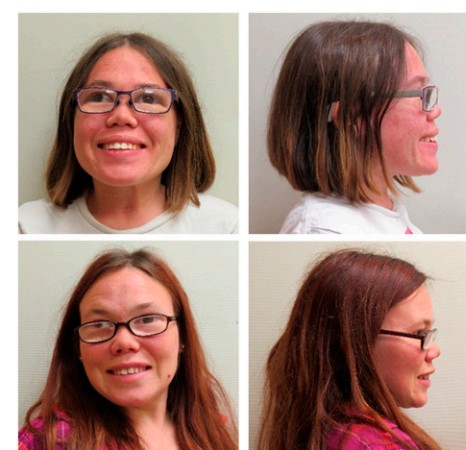

**B**

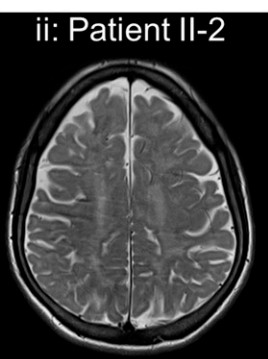

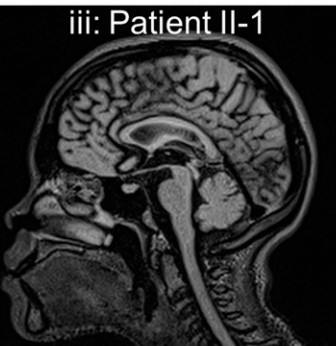

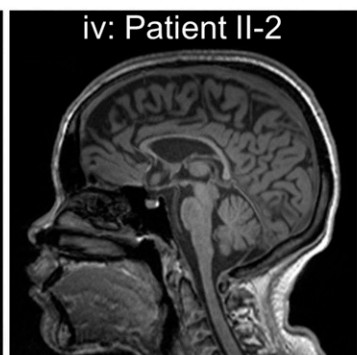

**C**

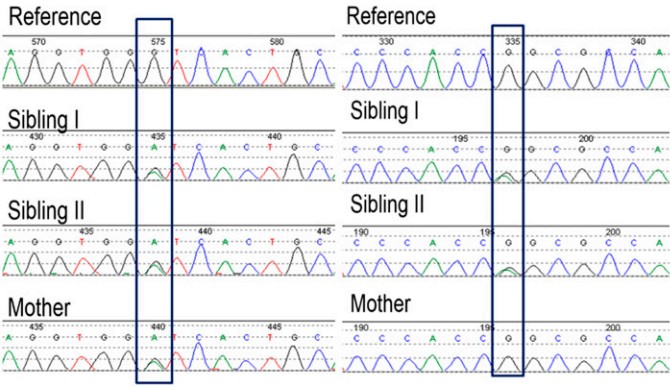

**D**

**Figure 1. Clinical and genetic patient data.**
**(A)** Pedigree of the patient family along with pictures of the two siblings in infancy and in adulthood. Note strabismus, midface hypoplasia, and depressed nasal ridge.
**(B)** Cranial MRI scans of the two sisters. Panes (i) and (iii) are individual II-1 at 22 y of age. Panes (ii) and (iv) are individual II- 2 at 25 y of age. Panes (i) and (ii) are axial T2-weighted images lacking the normal T2 hypointense signal, demonstrating hypomyelination. Panes (iii) and (iv) are sagittal MRI scans that revealed generalized hypomyelination of the corpus callosum. **(C)** Electropherogram conformation of the identified variants using Sanger sequencing, with mutated residues boxed.
**(D)** Sequence alignment of PISD homologs from the indicated species showing region containing the R277Q variant, with the arginine 277 residue highlighted in purple. A conserved histidine residue essential for autocatalysis is highlighted in green and one of four missense variants in a yeast Psd1p *temperature sensitive* allele is highlighted in yellow (Birner et al, 2003; Choi et al, 2015; Ogunbona et al, 2017).

(GERP score: 5.19) in exon 6 of *PISD* (Fig 1D). The variant had a CADD score of 35 and was predicted to be tolerated by SIFT (score: 0.19; median: 3.05), disease-causing by MutationTaster (*P*-value: 1) and probably damaging by PolyPhen (score of 0.920 [sensitivity: 0.68; specificity: 0.90]). The missense variant was reported with a low frequency of 0.01923% in the Genome

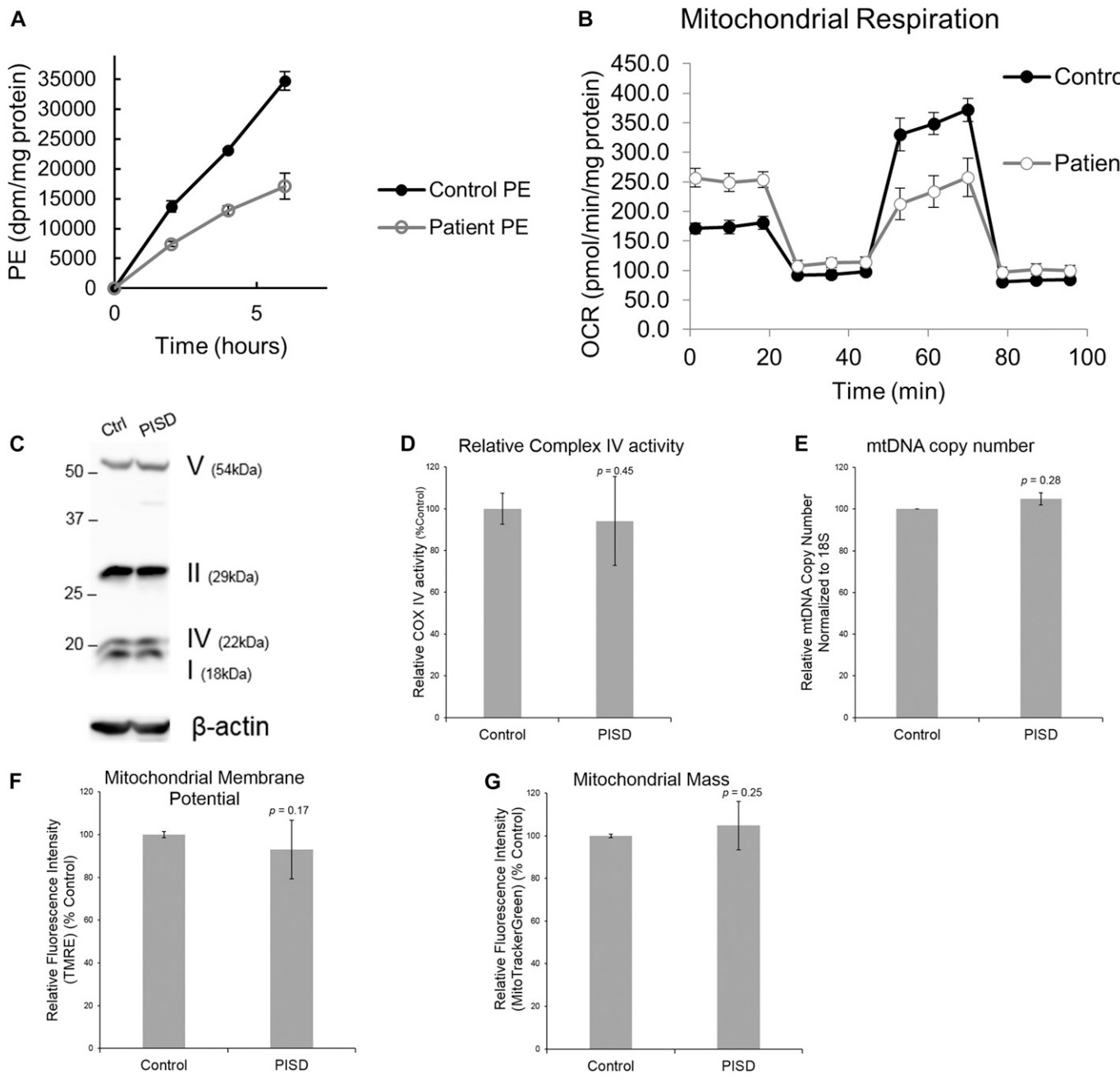

**Figure 2.  Characterization of PE synthesis and mitochondrial function in patient fibroblast cells.**
**(A)** Patient fibroblasts have decreased ability to convert PS into PE. Control and patient fibroblasts were labelled with [$^3$H] serine for 2, 4, and 6 h, after which total PE was isolated by thin-layer chromatography, and the incorporation rate of the [$^3$H] label into PE was quantified. **(B)** Patient fibroblasts have decreased maximal respiration. Profiles of oxygen consumption rate (OCR) for control and patient fibroblast cells, measured using the Seahorse Extracellular Flux XF24 Analyzer. Each point on the plot represents the average of four technical replicates, and the experiment has been replicated independently three times. Error bars represent SD. **(C–G)** There are no differences between control and patient fibroblasts with regard to the expression of mitochondrial OxPhos complex subunits in Western blots with $\beta$-ACTIN as a load control (C) (shown is a representative blot, and the experiment has been repeated independently three times), changes in Complex IV activity measured using a Dipstick Assay Kit (D), relative mtDNA copy number measured by qPCR (E), membrane potential measured by TMRE (F), or mitochondrial mass measured by Mitotracker Green (G). For panels (D–G), graphs represent the average from three independent biological repeats, each performed with three technical replicates. Error bars represent SD, whereas $P$-values were determined by unpaired two-tailed $t$ tests.

Aggregation Database (gnomAD), 0.03% for European American populations in the Exome Sequencing Project (ESP) database, and 0.02% in 1,000 genomes database (accessed on May 14, 2018). Meanwhile, the splice variant, c.697+5G>A, was not reported in

ClinVar, and no individuals were reported to be homozygous for the variant in gnomAD (accessed 14 May, 2018). The splice variant was predicted to weaken the splice donor site located at the end of exon 5 and was not reported in gnomAD, Exome Sequencing

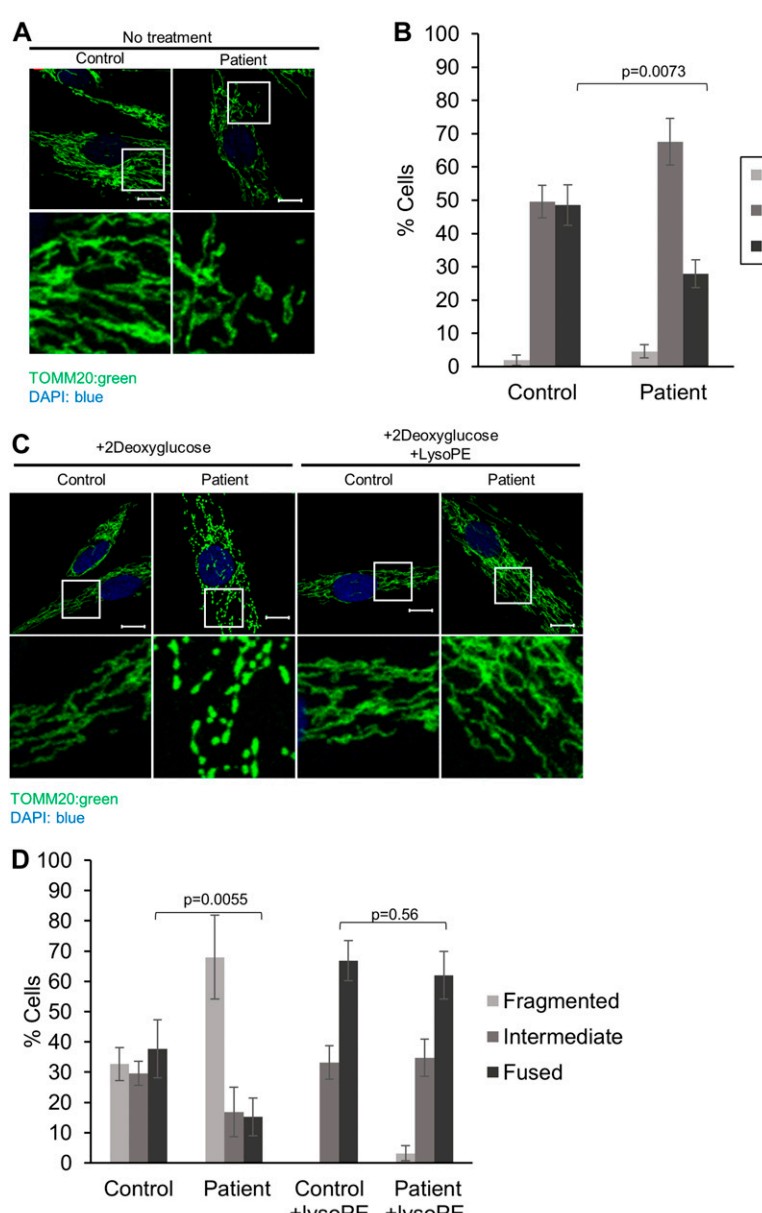

**Figure 3. Mitochondrial fragmentation in PISD patient fibroblasts is rescued by supplementation with lyso-PE.**
**(A)** Representative images of mitochondrial morphology under normal growth conditions, stained by immunofluorescence using a TOMM20 antibody (green), imaged with a Zeiss confocal microscope. Nuclei were stained with DAPI (blue). Bottom panels are a magnification of the white-boxed areas shown in the upper panel. Scale bars indicate 10 $\mu$m. **(B)** Quantification of mitochondrial morphology from cells in panel (A). **(C)** Representative images of mitochondrial morphology of cells treated with 2-deoxyglucose (20 mM) or lyso-PE (50 $\mu$M) for 48 h, as indicated. **(D)** Quantification of mitochondrial morphology of cells from in panel (C). For all statistical analysis, at least 50 cells from three technical replicates were quantified, the average percentage of the cell in each morphology category is plotted, and the experiment has been replicated independently three times. Error bars represent SD, and *P*-values were determined by unpaired two-tailed *t* tests compared with the number of fused cells in control.

Project, 1,000 genomes, or ClinVar databases (accessed on May 14, 2018).

## Fibroblast characterization

Given that there were no previous reports of individuals with pathogenic variants in *PISD*, combined with the unusual clinical presentation for a mitochondrial disease, we sought to perform additional functional analysis of patient fibroblast cells previously obtained from individual II-I to determine whether mitochondrial and/or PISD function were impaired. To begin, we measured the ability of both patient- and age-matched control fibroblast cells to convert PS to PE. We observed an approximately 50% decrease in PE

synthesis from PS in the patient fibroblast cells, consistent with impaired PISD activity (Fig 2A).

Next, we sought to investigate the effect that PISD deficiency might have on mitochondrial function. We observed marked differences between control and PISD patient fibroblasts using a Seahorse metabolic flux analyzer. Whereas basal respiration was elevated in PISD fibroblast cells, maximal respiration was decreased. In fact, PISD fibroblasts had no reserve capacity (Fig 2B). These findings clearly demonstrate alterations in the mitochondrial activity of PISD fibroblast cells. In an attempt to gain further mechanistic insight, we investigated additional parameters of mitochondrial function. As complex IV activity is reported to be decreased in CHO cells with severe knockdown of PISD expression, we examined the expression of several oxidative phosphorylation

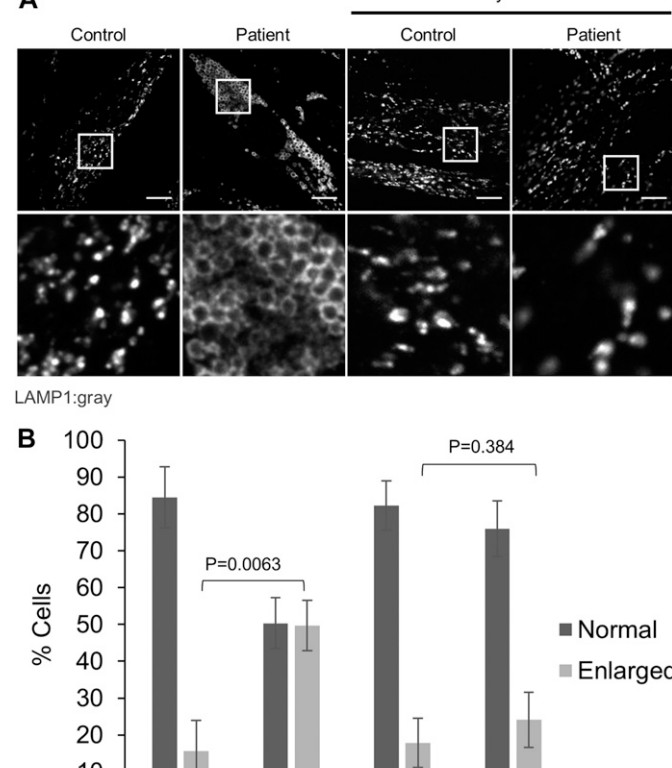

LAMP1:gray

**Figure 4. Altered lysosomal morphology in PISD patient fibroblasts is rescued by supplementation with lyso-PE.**
**(A)** Representative images of lysosomes stained by immunofluorescence (IF) against LAMP1 (gray) imaged with an Olympus SD-OSR confocal microscope. The cells were grown in either normal medium or medium supplemented with lyso-PE (50 $\mu$M) for 48 h, as indicated. Bottom panels are magnifications of the white-boxed areas in the upper panel. Scale bars indicate 10 $\mu$m. **(B)** Quantification of lysosomal morphology from three technical replicates, each counting at least 100 cells per condition, with the average percentage of cells with each morphology plotted. Error bars represent SD, and *P*-values were determined by unpaired two-tailed *t* tests compared with the number cells with enlarged lysosomes in control. Similar trends were replicated in three independent experiments.

proteins (Fig 2C) and more specifically activity of complex IV (Fig 2D). The fact that complex IV activity was unaffected in patient fibroblasts likely reflects a cell-type difference or milder deficiency in PISD activity compared with previous studies (Tasseva et al, 2013). Next, we examined mtDNA copy number, mitochondrial biogenesis, and mitochondrial membrane potential (Fig. 2E–G), but again found no significant differences.

To look at a more general indicator of mitochondrial function, we quantified mitochondrial morphology via confocal microscopy, as mitochondrial fragmentation is associated with mitochondrial dysfunction. In particular, PE is considered to be a pro-fusion lipid, and mitochondrial fragmentation has been reported in cells deficient in PISD activity (Steenbergen et al, 2005; Tasseva et al, 2013). A shift towards a more intermediate mitochondrial morphology was observed in patient fibroblast cells under normal growth conditions

(Fig 3A and B). We then treated cells with 2-deoxyglucose (2DG), which blocks glycolysis and forces cells to rely on mitochondrial function for their energy demands. Notably, this treatment has been used previously to exacerbate fragmentation in multiple mitochondrial disease patient fibroblasts (Guillery et al, 2008). Upon 2DG treatment, patient fibroblasts exhibited a much more dramatic fragmentation phenotype compared with control fibroblasts (Fig 3C and D).

We also examined lysosome structure, which can be impaired by mitochondrial dysfunction (Demers-Lamarche et al, 2016). We observed a marked increase in the number of PISD patient fibroblast cells exhibiting enlarged lysosomal structures compared with control fibroblast cells (Fig 4A and B). Taken together with the alterations in mitochondrial function and morphology, these findings provide strong evidence of mitochondrial dysfunction in PISD patient fibroblasts.

Finally, to link the mitochondrial dysfunction in PISD patient fibroblasts to deficient PISD activity, we treated cells with lyso-PE, which can replenish the mitochondrial pool of PE when PISD activity is impaired (Riekhof & Voelker, 2006; Riekhof et al, 2007; Tasseva et al, 2013). Although lyso-PE treatment promoted mitochondrial fusion to some degree even in control cells, we note that the extreme fragmentation observed in patient cells treated with 2DG was completely rescued via lyso-PE supplementation (Fig 3C and D). Moreover, lyso-PE treatment also restored lysosomal morphology (Fig 4A and B). These findings provide evidence that impaired mitochondrial morphology in the patient fibroblasts is due to low levels of PE in the cells, and that the mitochondrial dysfunction was causing secondary effects on lysosome structure.

### Genetic complementation

To confirm that impairments in PISD were the underlying cause of mitochondrial dysfunction in patient fibroblasts, we performed a genetic complementation by transfecting fibroblast cells with WT-*PISD* (Fig 5A). As transfection efficiency of the fibroblast cells was low (~5%), we focused on examining the previously described mitochondrial and lysosome morphology phenotypes in cells where we could confirm transfection by immunofluorescence. During the analysis, we noted that the cells expressing highest levels of PISD all exhibited fragmented mitochondrial networks, even in control fibroblasts. As this likely reflects an overexpression artifact, cells expressing high levels were excluded from further morphological analysis. Notably, we found that patient fibroblasts transfected with WT-*PISD* were indistinguishable from control fibroblasts with respect to both mitochondrial and lysosome morphology phenotypes (Fig 5B and C), indicating that similar to lyso-PE supplementation, expression of WT-PISD rescued the mitochondrial dysfunction.

### Functional characterization of pathogenic *PISD* variants

Additional functional analyses were performed to provide evidence that the two *PISD* variants in our patients were pathogenic. First, we examined the effect of the c.697+5G>A variant on splicing of the *PISD* mRNA. This variant located in intron 5, five base pairs after the end of exon 5, is predicted to weaken the endogenous

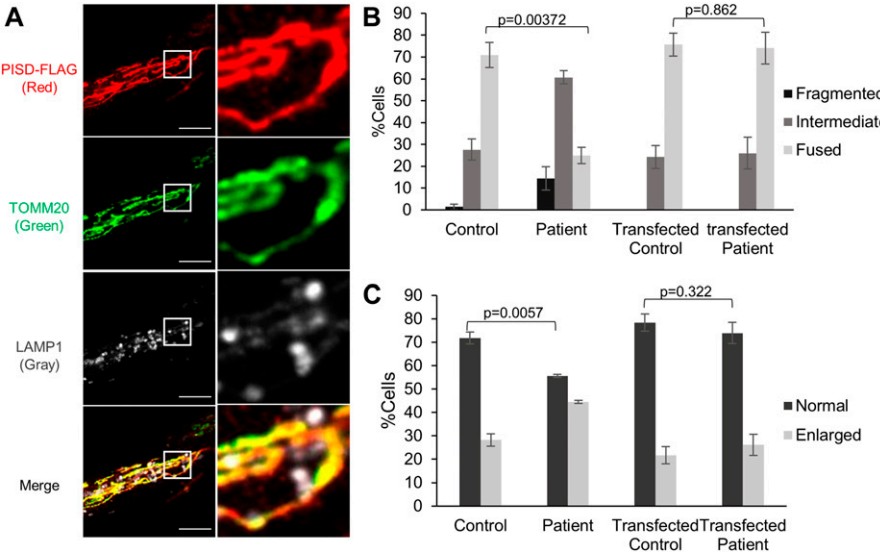

**Figure 5. Genetic complementation of PISD fibroblasts restores mitochondrial morphology in 2DG-treated cells.**
**(A)** Representative image of a patient fibroblast transfected with PISD-FLAG showing mitochondrial localization of PISD and normal morphology for both mitochondria and lysosomes, imaged with an Olympus SD-OSR confocal microscope. Mitochondria, PISD-FLAG, and lysosomes were stained by IF against TOMM20 (green), FLAG (red), and LAMP1 (gray), respectively. Scale bars indicate 5 μm. Right panels are magnifications of the boxed area. **(B)** Quantification of mitochondrial morphology for control and PISD patient fibroblasts either untransfected, or expressing PISD-FLAG, following 2DG treatment for 48 h. **(C)** Quantification of lysosome morphology for control and PISD patient fibroblasts either untransfected, or expressing PISD-FLAG. For both mitochondrial and lysosome morphology quantification, images from three independent experiments of at least 50 cells per condition were quantified, with the average percentage of cells with each morphology plotted. Error bars represent SD, and *P*-values were determined by unpaired two-tailed *t* tests compared with the morphology of control fibroblasts.

splice donor site at the end of exon 5, with a cryptic splice donor site located 76 base pairs upstream predicted to occur instead. The use of the upstream cryptic splice site would result in the skipping of the end of exon 5, with a 1-base pair frameshift for 51 amino acids and ending in a premature stop codon in exon 7. To investigate *PISD* splicing products, we generated cDNA from control and patient fibroblast RNA. PCR amplification across the region containing the splice sites allowed us to visualize an alternative splicing product in patient samples as a minor PCR band that was not present in control (Fig 6). We also note that treating cells with cycloheximide increased the abundance of the mis-spliced transcript, demonstrating that it is subject to nonsense-mediated decay. In addition, when performing quantitative RT-PCR across cryptic splice donor site, distinct peaks were observed in the melt

**Figure 6. Evidence for alternative splicing and nonsense mediated decay (NMD) of PISD mRNA in patient fibroblasts.**
cDNA generated from control and patient fibroblasts was amplified by PCR across the region containing the splice sites in question. The cells were treated with cycloheximide (CHX) (10 μg/ml for 4 h) to block nonsense-mediated decay, as indicated. The 310-bp product corresponds to the normal splice product. The smaller ~234-bp product corresponds to the predicted alternative splicing product, is visible in patient cDNA, and is more abundant following CHX treatment. Gene amplification of RPL13A was performed as a load control. Results were replicated in three independent experiments.

curves from control and patient samples (Fig S1A). Finally, a similar pattern of mis-splicing was observed in cDNA generated from RNA peripheral blood samples provided by patient II-2 and the mother (I-2), as a low-level signal in the sequencing electropherograms (Fig S1B). Together, these findings indicate that the c.697+5G>A variant does indeed alter splicing.

Next, we characterized the c.830G>A [p.Arg277Gln] variant, which was predicted to be pathogenic. To begin, we used the yeast *Saccharomyces cerevisiae*, an established and robust model for mitochondrial disease (Baile & Claypool, 2013). Previously, a protein sequence alignment of phosphatidylserine decarboxylase (PSD) enzymes from humans to bacteria was used to identify evolutionarily conserved residues that are essential for the autocatalytic proteolysis of yeast phosphatidylserine decarboxylase 1 (Psd1p) (Ogunbona et al, 2017). Interestingly, R358 in yeast Psd1p, corresponding to human R277, occurs two residues downstream of one of the four missense mutations present in a *temperature sensitive PSD1* allele (Figs 1D and 7A) (Birner et al, 2003). Furthermore, it is also in the vicinity of His345, a component of the conserved catalytic triad that is essential for Psd1p autocatalytic proteolysis (Ogunbona et al, 2017). As such, we hypothesized that the R358Q mutation may disrupt Psd1p autocatalytic proteolysis via a mechanism that is enhanced at elevated temperatures. Importantly, both the expression (Fig 7B) and autocatalytic processing (Fig 7C) of the R358Q mutant was similar to WT at 30°C. Consistent with our hypothesis, autocatalytic proteolysis of the R358Q mutant was significantly impaired at 37°C (Fig 7C). In contrast, self-processing of WT Psd1p into separate α and β subunits occurred at both tested temperatures. Although autocatalytic proteolysis was impaired for the R358Q mutant at elevated temperature, it was not completely ablated. Consistent with the persistence of some functional enzyme, the R358Q mutant enabled growth of the *psd1Δpsd2Δ* strain at 30°C and 37°C in the absence of ethanolamine (Fig 7D). These results establish the R358Q mutant as a novel *temperature-sensitive PSD1* allele with severely impaired, but not completely ablated, autocatalytic proteolysis at super-physiological temperatures.

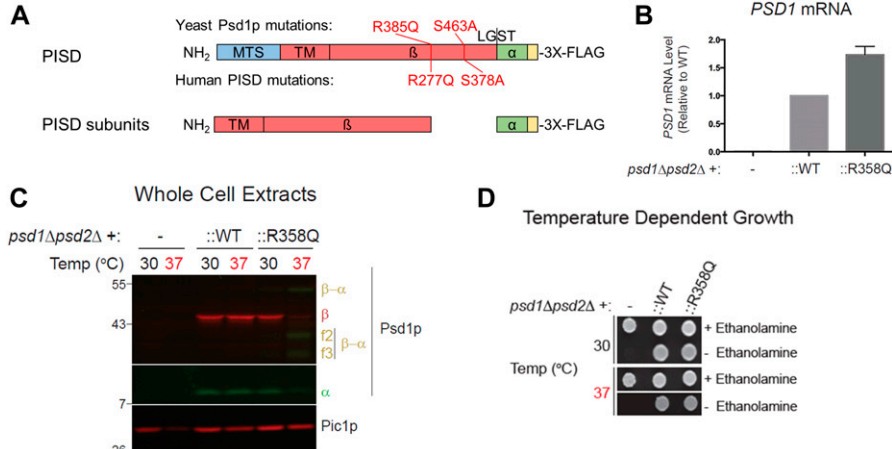

**Figure 7. The R277Q variant disturbs autocatalytic proteolysis when modeled in yeast Psd1p.**
**(A)** Schematic of the PISD and Psd1p constructs examined in this study, with numbering of variants indicated in red. A 3XFLAG tag was added to the C terminus of both constructs to enable detection of the α subunit following autocatalytic cleavage at the conserved LGST site necessary to generate the PISD proenzyme. **(B)** Relative *PSD1* mRNA transcript levels were determined by two-step reverse transcription–quantitative PCR. The CT values were normalized to the nuclear housekeeping gene *ACT1*, and fold change expressed relative to WT which was set at 1 (n = 4, SEM). **(C)** The α (anti-FLAG mouse monoclonal) and β (anti-Psd1p rabbit antisera) subunits of Psd1p were analyzed by immunoblotting in cell extracts isolated from cultures grown at the indicated temperatures; Pic1p served as a loading control. α-β indicates Psd1p that has not undergone autocatalytic proteolysis. f2 and f3 mark proteolytic fragments generated from the non-processed Psd1p precursor.

The migration of molecular mass markers in KiloDalton is indicated at the left. **(D)** Pre-cultures (30°C) of *psd1Δpsd2Δ* yeast, untransformed or transformed as indicated, were spotted onto synthetic complete dextrose plates with or without 2 mM ethanolamine and incubated at 30°C or 37°C for 3 d. Similar trends were replicated in three independent experiments. MTS, mitochondrial targeting signal; TM, transmembrane domain.

These findings in yeast prompted us to investigate the autocatalytic proteolysis of the human PISD protein containing the R277Q variant (Fig 8A). However, our PISD antibody was unable to detect the endogenous PISD protein in either control or patient fibroblast cells. Thus, we turned to an overexpression approach, where we expressed WT or mutant PISD containing a C-terminal FLAG tag in HEK cells. Following a 24-h transient transfection of the wild-type PISD, we detected both the 12-kD α and 30-kD β PISD subunits with our PISD antibody and a FLAG antibody, respectively. However, we could not detect any of the R277Q mutant protein 24 h after transfection (Fig S2A). Thus, we looked at protein levels 96 h following transfection (Fig 8). In addition to the α and β subunits, we also detected a significant accumulation of the unprocessed 45-kD precursor for the wild-type PISD at 96 h. In contrast, when the R277Q mutant was overexpressed, only the 45-kD PISD precursor was observed, indicating that the protein was indeed translated. Importantly, quantitative RT-PCR confirmed that similar levels of PISD mRNA were expressed from the plasmid constructs 96 h post transfection (Fig S2B). Thus, similar to experiments in yeast, we note that the R277Q variant severely impaired the autocatalytic processing of the human PISD protein. As a control, we also generated a *PISD* construct containing the S378A variant, which corresponds to the yeast Psd1 S463A mutation that also has impaired autocatalytic processing. Similar to the patient R277Q variant, when the S378A variant was overexpressed, the unprocessed 45-kD PISD precursor was the predominant form. Finally, we expressed the nearby C266Y variant in PISD, which was recently reported in patients with similar phenotypes to the sisters described herein (Girisha et al, 2018). Although low levels of both α and β subunits were detectable, we again observed reduced autocatalytic processing of PISD, consistent with a similar underlying mechanism for these pathogenic variants.

### PISD and alterations to mitochondrial protein homeostasis

Finally, given that the phenotypic characteristics of our PISD patients resembled those of mitochondrial chaperonopathies, we investigated whether or not mitochondrial proteases as well as mitochondrial protein homeostasis were impaired in PISD patient fibroblast cells treated with 2DG, where we saw dramatic changes in mitochondrial morphology. Given the importance of PE in the IMM, we started with OMA1, a zinc metalloprotease that is activated upon mitochondrial stress (Levytskyy et al, 2017). We found that the levels of OMA1 protein expression were severely decreased in PISD patient fibroblast cells (Fig 9A). To determine whether this depletion was

**Figure 8. Impaired autocatalytic proteolysis of R277Q and C266Y pathogenic PISD variants.**
Western blot analysis of PISD fragments in HEK cells overexpressing the indicated PISD constructs for 96 h. Wild-type PISD undergoes autocatalytic cleavage into 30-kD β-subunits and 12-kD α-subunits required to form an active PISD enzyme. The unprocessed 45-kD PISD protein is the predominant protein form for the PISD mutant constructs. Results were replicated in three independent experiments.

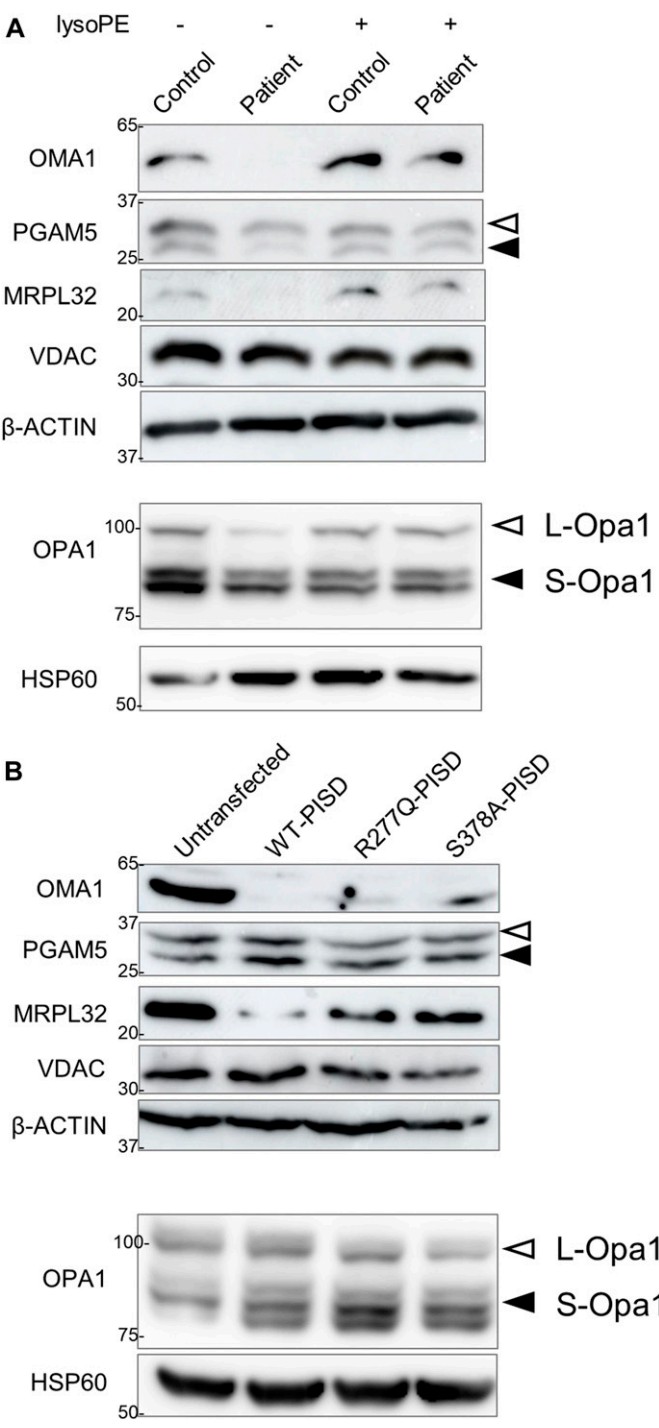

**Figure 9. Altered mitochondrial protein homeostasis in PISD patient fibroblasts.**
**(A)** Western blot analysis of various mitochondrial proteins in control and patient fibroblasts treated with 2-deoxyglucose (20 mM) and 50 $\mu$M lyso-PE for 48 h as indicated. VDAC was used as a marker for total mitochondrial signal, whereas $\beta$-ACTIN was used as a general load control. Open and solid triangles indicate unprocessed and processed forms, respectively. Decreased levels of OMA1, OPA1, MRPL32, and processed PGAM5 were observed in patient fibroblast cells, but rescued upon lyso-PE treatment. To obtain optimal separation of OPA1 bands, separate gels were run and blotted with HSP60 as a load control. **(B)** Protein extracts from HEK cells overexpressing wild-type or mutant PISD constructs (as in Fig 8) were analyzed by Western blot for the same proteins as in part (A).

sufficient to alter protein homeostasis, we examined the cleavage PGAM5, which is processed from a long to a short form by several membrane-bound IMM proteases, such as OMA1 (Wai et al, 2016) and PARL (Sekine et al, 2012). In particular, we found decreased levels of the short form of the PGAM5 protein in patient fibroblast cells, suggestive of decreased IMM protease activity. To look for more global effects on mitochondrial IMM proteases, we also examined other established targets of mitochondrial IMM proteases. We saw changes to the levels of OPA1, an IMM protein that regulates mitochondrial fusion, and which is regulated by multiple proteases (MacVicar & Langer, 2016), including OMA1 (Ehses et al, 2009; Head et al, 2009). In addition, we saw decreased levels of the mature form of MRPL32, a mitochondrial ribosomal protein subunit that is processed by the IMM $m$-AAA protease (Nolden et al, 2005). Finally, we found that treating PISD patient fibroblasts with lyso-PE rescued the changes in OMA1, PGAM5, OPA1, and MRPL32 proteins. Importantly, quantitative RT-PCR revealed that changes in mRNA levels cannot explain all the differences in protein stability between control and patient fibroblasts, nor upon rescue by lyso-PE (Fig S3A), indicating that global changes in these protein levels are post-translational in nature.

We also examined the effects of overexpressing wild-type and mutant PISD on mitochondrial protein homeostasis in HEK cells (Fig 9B). Notably, overexpressing WT-PISD alone significantly reduced the levels of OMA1 and MRPL32, but did not have any major effects on PGAM5 or OPA1. Meanwhile, overexpressing the R277Q and S378A PISD mutants also decreased levels of OMA1 and MRPL32 proteins, although to a less extent than WT-PISD. Again, quantitative RT-PCR showed that differences in mRNA expression cannot account for all the changes in protein expression in cells overexpressing the various PISD constructs (Fig S3B). Altogether, these data are consistent with the notion that altering levels of PE in the IMM, either up or down, impacts mitochondrial proteases and alters mitochondrial protein homeostasis.

## Discussion

Consistent with impaired PISD activity, our characterization of PISD patient fibroblasts revealed impaired phospholipid metabolism, altered mitochondrial respiration, and fragmentation of the mitochondrial network, which was exacerbated by 2DG. The fact that supplementation of patient fibroblasts with lyso-PE and genetic complementation with WT-PISD rescued mitochondrial fragmentation in 2DG-treated patient cells strongly supports a mitochondrial PE deficiency due to decreased PISD activity in these cells. Furthermore, functional characterization of the PISD patient variants shows the presence of an alternative splicing event of the maternally encoded transcript and reduced autocatalytic proteolysis of the paternally encoded protein, demonstrating that these polymorphisms reduce PISD activity and are pathogenic.

Overexpression of wild-type PISD leads to a dramatic decreased in OMA1 and MRPL32, which is blunted when mutant PISD proteins are overexpressed. Similar results were replicated in three independent experiments.

An initial quandary when PISD was identified as the only candidate gene was why the patient phenotype did not resemble classic mitochondrial disease. To this end, the fact that we did not observe any significant differences for several other mitochondrial parameters, suggests that the underlying mitochondrial dysfunction is different from what is typically found in classic mitochondrial diseases. Importantly, during the revision of this manuscript, an article was published describing the C266Y pathogenic variant in PISD in patients with similar phenotypes (Girisha et al, 2018), further supporting PISD as the responsible gene. The similarity of skeletal phenotypes in PISD patients with recently described skeletal abnormalities in mitochondrial chaperonopathies was notable. To this end, our data suggesting that IMM proteases and mitochondrial protein homeostasis are impaired in PISD patient fibroblasts may explain the similar patient phenotypes.

We considered two possible mechanisms by which pathogenic variants in PISD might impair mitochondrial protein homeostasis. The first possibility was that accumulation of the unprocessed PISD protein reduced the ability of mitochondrial proteases to maintain normal mitochondrial proteostasis. To this point, it is likely that unprocessed human PISD is degraded by mitochondrial proteases, similar to the situation in yeast, where the unprocessed autocatalytic mutant Psd1 protein is degraded by the OMA1 protease (Ogunbona et al, 2017). In this regard, the fact that the unprocessed PISD protein was only visible 96 h post transfection is consistent with an inability of mitochondrial proteases to efficiently remove the accumulated 45-kD precursor. However, given that transcript levels were increased by several orders of magnitude in HEK cells, we are hesitant to draw conclusions on whether normal levels of the mutant PISD protein would be sufficient to impair mitochondrial protein homeostasis and lead to pathology. Moreover, given that the unaffected father presumably carries a single copy of R277Q, it is unlikely that the presence of the mutant protein alone, at physiological levels, is sufficient to induce pathology. However, as DNA from the father was unavailable, we could not completely exclude the unlikely possibility of gonadal mosaicism.

The second, more likely possibility is that the global reduction of PISD activity and subsequent decrease in PE levels contributes indirectly to impaired mitochondrial protein homeostasis. The fact that depletion of PISD leads to mitochondrial fragmentation and decreased oxidative phosphorylation (Steenbergen et al, 2005; Tasseva et al, 2013) demonstrates that depletion of mitochondrial PE is sufficient to cause cellular defects. To this end, several mitochondrial proteases are associated with, or are integral to, the IMM (e.g., HTRA2/OMI, iAAA, mAAA, OMA1, and PARL), and their functions can be regulated by membrane lipid domains (Quiros et al, 2015). As such, these proteases are likely sensitive to alterations in membrane lipid composition such as decreased levels of PE.

The relationship among mitochondrial proteases is complex, as they often cleave and activate each other (Levytskyy et al, 2017), and it is not possible to directly measure protease activity. Nonetheless, our Western analysis of mitochondrial proteins in PISD patient fibroblasts provide strong evidence for decreased mitochondrial protease activity and subsequent impairments in mitochondrial protein homeostasis. In particular, we observed a marked decrease

in the protein levels of the protease OMA1, but not its mRNA levels. This decrease in OMA1 is likely due to destabilization of the protein itself because of altered lipid composition and/or increased degradation of OMA1 by other mitochondrial proteases that cleave OMA1. Notably, $oma1^{-/-}$ mice are viable, and although they have an obesity phenotype and defective thermogenesis (Quiros et al, 2012), it does not match the phenotypes of our PISD patients. This dissimilarity in phenotypes suggests that loss of OMA1 alone is not sufficient to induce the pathology seen in our PISD patients, where there must be additional contributing factors. Certainly, our data point to the fact that multiple IMM proteases are affected in PISD fibroblasts, as we also see differences in OPA1, PGAM5, and MRPL32 protein profiles, that cannot all be explained by changes to mRNA. Importantly, the rescue of these protein profiles in patient fibroblasts supplemented with lyso-PE demonstrates that the altered mitochondrial protein homeostasis is due to depleted PE levels. Finally, the fact that we also see changes to mitochondrial protein homeostasis when PISD is overexpressed in HEK cells, albeit with a different profile than the changes in PISD fibroblasts, demonstrates that higher levels of PE can also be detrimental. This finding is also supported by the mitochondrial fragmentation we observed in fibroblast cells that highly expressed WT-PISD. The different mitochondrial protein profiles in patient fibroblasts deficient in PE and HEK cells overexpressing PISD likely reflects that the various IMM proteases respond differently to either an excess or deficiency of PE levels. Altogether, our data support the notion that IMM proteases are sensitive to changes in PE levels.

Another open question pertains to the loss of PISD activity required for pathogenesis. The fact that both parents were unaffected, combined with the fact PISD +/− mice appear normal (Steenbergen et al, 2005), suggests that the threshold of PISD depletion required for pathogenicity lies somewhere between 50 and 100%. In this light, it is likely that the PISD patients still produce some functional PISD protein, as PISD knockout mice are embryonic lethal (Steenbergen et al, 2005). Although we cannot currently assign the relative contribution of each patient allele to the remaining PISD function in patients, it is notable that the R358Q mutation in yeast Psd1 retains some activity. Similarly, we also see low-level autocatalytic processing of the recently described C266Y variant. Finally, we note that overexpression of the R277Q and S378A mutants in HEK cells led to a milder depletion of OMA1 and MRPL32 compared with control, consistent with reduced activity. Moreover, it is also possible that a portion of the maternal allele is spliced normally, or that the amount of alternative splicing varies in different tissue types.

Collectively, our data demonstrate that pathogenic variants in PISD lead to mitochondrial dysfunction that likely leads to the unique array of patient phenotypes and demonstrate that PISD is a novel human disease gene. Although cataracts are often observed in classic mitochondrial disease, the severe growth impairment with nonspecific skeletal anomalies and dysmorphic features are not. However, the combination of skeletal abnormalities with cataracts, distinctive craniofacial features, including depressed nasal ridge and intellectual disability are all reminiscent of patient phenotypes described in the recently described subclass of mitochondrial chaperonopathies. Our data suggest that impaired

mitochondrial protein homeostasis as a result of pathogenic *PISD* variants may explain the similarities of the PISD patient phenotype to those of mitochondrial chaperonopathies. This intriguing finding may help define a better mechanistic understanding of mitochondrial chaperonopathies. Furthermore, this work adds to the broad phenotypic spectrum in mitochondrial phospholipid metabolism-based diseases (Lu & Claypool, 2015).

# Materials and Methods

## Patient selection and clinical investigations

The two patients, of Caucasian/European ancestry, were ascertained through the clinical practice of one of the authors at the Alberta Children's Hospital. Consent for participation and for the publication of patient photos from both patients and their mother was obtained as part of a study approved by the Conjoint Ethics Review Board of the University of Calgary. Consent for experiments conformed to the principles of the WMA Declaration of Helsinki and the Department for Health and Human Services Belmont Report. Investigations such as basic genetic and metabolic blood work, skeletal surveys, and MRI scans on both patients had been obtained during the course of their routine clinical care. As they were undiagnosed after conventional investigations, they were enrolled in a whole-exome sequencing–based research study to identify the cause of their presumed genetic rare disease.

## Molecular genetics studies

Genomic DNA (gDNA) was extracted from peripheral blood provided by the patients and their unaffected mother using an Autopure LS system and Gentra Puregene blood kit (QIAGEN). The Agilent SureSelect V5 All Exon kit (Agilent) and Illumina HiSeq 2500 were used for target enrichment and sequencing, respectively. Read alignment to the hg19 reference genome, as well as variant calling and annotation, were completed as described in previously published Care4Rare Consortium projects (Beaulieu et al, 2014).

Because of the rarity of the sisters' phenotype, annotated variants with an ExAC minor allele frequency less than 1% and seen in less than five previous Care4Rare samples were considered. An autosomal recessive mode of inheritance was assumed and included analysis of rare homozygous variants shared by both sisters and carried by the mother, or, rare bi-allelic variants shared by the sisters, of which only one of the variants was carried by the mother. Resulting rare variants, and their respective genes were visualized using Alamut Visual version 2.9.0 (Interactive Biosoftware). Conservation, in silico pathogenicity prediction scores (Polyphen, SIFT and MutationTaster), and splice prediction scores (SpliceSiteFinder-like, MaxEntScan, NNSPLICE, GeneSplicer and Human Splicing Finder) were taken in account. Sanger sequencing, using the ABI BigDye Terminator Cycle Sequencing kit v1.1 (Life Technologies) on a 3130xl genetic analyzer (Life Technologies), confirmed the candidate variants.

## cDNA sequencing and analysis

A PAXgene Blood RNA kit (PreAnalytiX) was used to extract RNA from peripheral blood samples provided by patient II-2 and their mother. cDNA was synthesized (SuperScript III; Invitrogen) from the extracted RNA. PCR amplification (HotStar Taq Plus; QIAGEN) across exon junctions of cDNA was performed using primers 5′-GGCG-AAATGGTTGCACTTC-3′ (forward) and 5′-GCAGGTCCCGGTCAAAG-3′ (reverse). Patient cDNA was Sanger-sequenced using the 5′- GGC-GAAATGGTTGCACTTC-3′ (forward) and 5′-GCAGGTCCCGGTCAAAG-3′ (reverse) primers as well as a sequencing primer 5′-GATGGAGG-CCCGTAAGC-3′ (forward) to ensure complete capture of exons 5 through eight of *PISD* (Ref Seq NM_014338.3). For mRNA expression quantification, total RNA was extracted from cell pellets using E.Z.N.A. Total RNA Kit (Omega) according to the manufacture's protocols. cDNA was synthesized using iScript Advanced cDNA Synthesis kit (Bio-Rad). Q-RT-PCR was performed using 20 $\mu$l reactions of PowerUp SYBR Green Master Mix (A25742; Thermo Fisher Scientific) with QuantStudio 6 Flex Real-Time PCR System (Thermo Fisher Scientific) according to the fast cycling mode protocol from the manufacturer. 50 ng of cDNA and 500 nM each of forward and reverse primers (final concentrations) were used. Primers used for the reaction: OMA1 forward 5′-CATTGTAGGCAGGGGCATAA-3′, OMA1 reverse 5′-CACCA-CAAAGAGCAATCCAAAA-3′, OPA1 forward 5′-TCAAGAAAAACTTGATG-CTTTCA-3′, OPA1 reverse 5′-GCAGAGCTGATTATGAGTACGATT-3′, PGAM5 forward 5′-TCGTCCATTCGTCTATGACGC-3′, PGAM5 reverse 5′-GGCTTC-CAATGAGACACGG-3′, MRPL32 forward 5′-TGTCCTTTGTGCCTACTGCTA-3′, MRPL32 reverse 5′-CTTGTTCAGACGGTGTCTCTC-3′, GAPDH forward 5′-TTGGTATCGTGGAAGGACTC-3′, GAPDH reverse 5′-ACAGTCTTCTGGGT-GGCAGT-3′, HPRT forward 5′-CTTTGCTGACCTGCTGGATT-3′, HPRT reverse 5′-TCCCCTGTTGACTGGTCATT-3′.

## Fibroblast cell culture growth conditions and treatments

Fibroblast cells were obtained from patient- and gender-/age-matched control from skin punch biopsies. The skin biopsy from individual II-I was taken at the time of her hospitalization at age 10 while undergoing an operative procedure. Fibroblast cells were cultured in MEM, without glutamine (Gibco) but supplemented with 2 mM of L-glutamine (Gibco) and 10% heat-inactivated FBS (Gibco). Fibroblast cells were detached from 10-cm culture dishes (Fisher) using Trypsin–EDTA (0.25%) and passed in a 1:3 ratio three times per week. For confocal microscopy experiments, fibroblast cells were seeded at 20,000 cells per well using 24-well plates on glass cover slips (12-545-81; Thermo Fisher Scientific) overnight. Fibroblast cells were treated with 50 $\mu$M lyso-PE (89576-29-4; Sigma-Aldrich) in ethanol and/or 20 mM 2-deoxyglucose (sc-202010; Santa Cruz) for 48 h.

## Genetic complementation of fibroblast cells

Fibroblast cells cultured to 70–80% confluency were harvested using Trypsin–EDTA (0.25%) and washed with 1× DPBS (Corning 20-031-CV). One million cells were suspended in 100 $\mu$l Opti-MEM (Gibco) with 20% heat-inactivated FBS (Gibco). Electroporation was performed using a Nucleofector II and the A-024 program. Transfected cells were then transferred immediately to cell culture

plates with culture media. The media were replaced after 4 h. For mitochondrial morphology, 20 h after electroporation, the cells were treated with 20 mM 2-deoxyglucose for an additional 48 h before fixation for imaging. For lysosome morphology, the cells were fixed for imaging 68 h after electroporation.

## HEK cell culture and transfection

HEK cells were cultured in DMEM (Gibco) with 10% heat-inactivated FBS (Gibco). Cells were seeded at $2 \times 10^6$ cells per plate on 10-cm plates overnight the night before transfection. Cells were transfected using Lipofectamine 3000 (Thermo Fisher Scientific) reagent with 15 $\mu$g plasmid DNA per plate. The human *PISD* ORF corresponding to isoform A (NCBI Reference Sequence [NP_001313340.1]) was amplified by PCR using cDNA from HeLa cells, cloned initially into pSP64 (Promega), and then subcloned into pcDNA5/FRT (Invitrogen) with a 3XFLAG tag appended to its carboxy terminus by overlap extension enabling detection of the $\alpha$ subunit. The R277Q and S378A mutant variants were generated by overlap extension PCR.

## Immunofluorescence

Cover slips were fixed in phosphate-buffered saline buffer containing 4% paraformaldehyde at 37°C for 15 min followed by quenching with ammonium chloride (50 $\mu$M). Fixed cover slips were stained with an $\alpha$TOMM20 antibody (sc-11415; Santa Cruz) for detection of mitochondria or an $\alpha$LAMP1 antibody (sc-18821; Santa Cruz) for detection of lysosomes. The samples were then incubated with appropriate Alexa Fluor–labelled secondary antibodies (Invitrogen). Finally, the cover slips were mounted for imaging with Dako fluorescent mounting medium containing DAPI (S3023; Agilent Technology).

## Mitochondrial morphology

Mitochondrial morphology was imaged using 488-nm lasers with either a 40× lens on a ZEISS LSM 700 confocal laser-scanning microscope running ZEN (blue edition) software or a 100× lens on an Olympus SD-OSR spinning-disc confocal microscope running MetaMorph software, as indicated in the figure legends. Quantification of mitochondrial morphology was performed blinded, by grading three replicates of at least 50 cells each, into three levels of mitochondrial morphology: fused (mitochondria form a mesh-like network with no short fragments), intermediate (mitochondria form a mixture of short networks and long fragments), and fragmented (most mitochondria form very short fragments with very little connectivity) according to the representative images of each level (Fig S4A). For statistical analyses, multiple technical or biological replicates were analyzed by unpaired two-tailed *t* tests (as indicated in the figure legends). For every experiment, at least three biological replicates were performed to confirm the reproducibility of the findings. For the genetic complementation studies, all cells with high levels of PISD expression were excluded from the analysis. High levels were defined as a pixel intensity over 2,000 for the PISD-FLAG channel, as determined quantitatively using the Image/Threshold function in ImageJ. This threshold was established by

the fact that all control cells expressing PISD above this level had highly fragmented mitochondria (Fig S4C), which were not present in untransfected cells, and thus deemed to be an artifact of overexpression.

## Lysosomal morphology

Lysosomal morphology was imaged using 568-nm lasers with 100x lens on an Olympus SD-OSR imaging system. Quantification of lysosomal morphology was performed blinded by scoring three replicates of 100 cells each, as either normal (lysosomes appear as distinct unconnected puncta) or enlarged (lysosomes appear larger, and are connected in sheet-like structures) according to representative images of each level (Fig S4B). For statistical analyses, multiple technical or biological replicates were analyzed by unpaired two-tailed *t* tests (as indicated in the figure legends). For every experiment, at least three biological replicates were performed to confirm the reproducibility of the findings.

## Oxygen consumption and respiratory function

Fibroblasts were seeded at 40,000 cells per well and grown overnight in XF24 plates (Agilent). On the day of the experiment, normal growth medium was replaced by Seahorse XF Base Medium containing 1 mM pyruvate, 2 mM glutamine, and 10 mM glucose, pH 7.4. After 45-min incubation at 37°C without $CO_2$, the cells were loaded into the Seahorse XF24 Analyzer for calibration. Oxygen consumption rate (OCR) was measured at three time points following serial incubations with oligomycin (final concentration 0.5 $\mu$M), FCCP (final concentration 1.0 $\mu$M) and antimycin (final concentration 0.5 $\mu$M). At the end of the experiment, the cells were lysed with radioimmunoprecipitation assay buffer, and total protein was determined by bicinchoninic acid assay. OCR was normalized to total protein/well. Analysis was performed on five technical replicates and confirmed in two biological replicates.

## Complex IV activity assay

The Complex IV human enzyme activity Dipstick Assay Kit (ab109876; MitoSciences) was used to assess cytochrome C oxidase activity in control and PISD fibroblasts according to the manufacturer's instructions. The assay was performed using 50 $\mu$g lysates. Following signal development, dipsticks were imaged on an Amersham Imager (AI600) and signal intensity, corresponding with complex IV activity, was quantified using ImageJ software (Schindelin et al, 2012). Values were determined from the average of at least three technical replicates from three independent biological replicates.

## Western blotting

Total protein lysates were prepared using radioimmunoprecipitation assay buffer. Protein concentration was determined by bicinchoninic acid assay. Protein (50 $\mu$g) from whole cell lysate was boiled with Laemmli sample buffer (Bio-Rad) and loaded into each lane of 12% SDS–PAGE gels then transferred to PVDF membranes. The membranes were then blocked with 5% skim milk and blotted with various primary antibodies: anti-$\beta$-ACTIN (A5316;

Sigma-Aldrich), anti-HSP60 (12165; Cell Signaling), anti-PISD (Custom made), anti-FLAG (NB600-344; Novus Biological), anti-MRPL32 (C14071; Assay Biotech), anti-OMA1 (ab104316; Abcam), anti-OPA1 (612606; BD Bioscience), anti-PGAM5 (HPA036978; Sigma-Aldrich), OXPHOS antibody cocktail (ab110411; Abcam), and anti-VDAC (ab14734; Abcam). All primary antibodies were used in 1:1,000 dilution. Appropriate secondary HRP-conjugated antibodies (Thermo Fisher Scientific) were used in 1:10,000 dilution. Custom-made monoclonal antibodies against human PISD were produced by ABclonal Science Inc. using purified recombinant protein as antigen (details below). A total of five hybridomas (PISD-1-3-4-2, PISD-1-6-1-2, PISD-2-3-3-3, PISD-2-6-2-1, and PISD-2-9-2-1) were established that secreted PISD reactive antibodies. Protein G–purified monoclonal antibodies from PISD-2-9-2-1 were used in this study. At least three biological replicates were performed to confirm the findings, with representative blots shown in figures.

### mtDNA copy number analysis

gDNA from control and patient fibroblasts was extracted using E.Z.N.A. Tissue DNA Extraction Systems, (D3396; Omega Bio-tek) according to the manufacturer's instructions. Relative mtDNA copy number was examined by real-time quantitative PCR (qPCR) using the QuantStudio 6 Flex Real-Time PCR system (Thermo Fisher Scientific). Primer sequences to amplify mtDNA and the nuclear-encoded housekeeping gene 18S, and thermocycling conditions were exactly as described in (Eaton et al, 2007). Analyses were performed in 20 $\mu$L reactions containing 10 $\mu$L PowerUp SYBR Green Master Mix (A25742; Thermo Fisher Scientific), 100 ng gDNA, and 500 nM forward and 500 nM reverse primers (final concentrations). MtDNA copy number relative to 18S was analyzed using the delta delta Ct method and presented as % control (Livak & Schmittgen, 2001). Copy number analyses were performed with the average of three technical replicates from each of three independent biological replicates.

### Mitochondrial membrane potential and mitochondrial mass analyses

Control and patient fibroblasts were seeded in six-well plates at 100,000 cells per well and allowed to grow for two days. On the day of analysis, the cells were incubated with the membrane potential–dependent dye TMRE (Tetramethylrhodamine, ethyl ester) (50 nM, 20 min) (T669; Life Technologies) to examine mitochondrial membrane potential. The ionophore FCCP (carbonylcyanide-4-(trifluoromethoxy)-phenylhydrazone, 10 $\mu$M) (BML-CM120; Enzo Life Sciences) was used as a negative control. In addition, another set of cells were labelled with the membrane potential–independent dye MitoTracker Green (50 nM, 20 min) (M7514; Life Technologies) to assess mitochondrial mass. Following staining, the cells were washed three times with pre-warmed PBS, trypsinized, and pelleted. The cell pellets were then resuspended in complete media, and signal intensity from the respective dyes was analyzed by flow cytometry using the BD LSR II flow cytometer (BD Biosciences) and the BD FACSDiva software. Mean fluorescence intensity was recorded for 20,000 events for triplicates of each condition, and the data were presented as percent control. Mitochondrial membrane potential

and mitochondrial mass analyses were performed on at least three independent replicates.

### Purification of recombinant human PISD

The predicted mature human PISD spanning residues 77-409 was cloned downstream of the 6× His tag provided by the pET28a vector (Novagen). Transformed BL21(DE3) pLysS RIL *Escherichia coli* cells were used to produce recombinant His$_6$PISD. A 1-liter culture of cells was induced at 37°C with 1 mM IPTG for 4 h and the cell pellet collected by centrifugation at 3,020 $g$ for 10 min. The cell pellet was resuspended in 250 ml of 0.9% NaCl solution, centrifuged at 3,020 $g$ for 10 min, and the supernatant again discarded. The cell pellet was resuspended in 40 ml of lysis buffer (50 mM NaH$_2$PO$_4$, 300 mM NaCl, 10 mM imidazole, 1% Tween-20, and 0.1 mM EDTA, pH 8.0), and the cells were incubated with 40 mg/ml lysozyme for 30 min on ice. Cell lysis was accomplished using an Avestin homogenizer. The cell lysate was centrifuged at 10,000 $g$ at 4°C for 20 min, and the cell pellet resuspended in inclusion-body solubilization buffer (1.67% Sarkosyl, 0.1 mM EDTA, 10 mM dithiothreitol, 10 mM Tris–HCl, and 0.05% PEG, pH 7.4), followed by 20-min incubation on ice. The suspension was mixed with 14 ml of 10 mM Tris–HCL (pH 7.4), centrifuged at 12,000 $g$ at 4°C for 10 min, after which the pellet was discarded. Protein in the supernatant was purified by passing the suspension through a Ni-NTA column (QIAGEN) and eluted with elution buffer (250 mM imidazole, 0.1% Sarkosyl, 50 mM NaH$_2$PO$_4$, 300 mM NaCl, and 10% glycerol, pH 8.0). The purified protein was run on a 12% SDS–PAGE gel, then further isolated from contaminating proteins by removing a gel slice corresponding to 41 kD and incubating it with 50 ml 1× PBS at 37°C overnight. The protein was then concentrated by centrifugation with an Amicon Ultra-4 (Millipore) filter at 4,000 $g$ for 10 min. The resulting 4.5 ml of purified protein was dialyzed overnight in 0.5% SDS in PBS at room temperature to remove excess SDS. This resulted in 8.3 mg of His$_6$PISD in 4.5 ml.

### Confirmation of alternative splicing variant

Total cellular RNA was isolated using total RNA kit (Omega) according to the manufacturer's instructions. cDNA was synthesized by iScript advanced cDNA kit (Bio-Rad) according to the manufacturer's instructions. PCR amplification of the PISD splice variant was performed using Taq polymerase (Invitrogen) at 58°C for 45 cycles on Bio-Rad S1000 thermal cycler. Template DNA (300 ng) was used with 10 $\mu$M of forward and reverse primers. Forward primer 5′-GCCTGCACAGCG-TGATTAG-3′ and reverse primer 5′-TGACATCAGGGAGCCTGG-3′ were chosen at exon junctions to eliminate genomic DNA amplification. Resulting PCR product was visualized with EZ vision dye loading buffer (Life Technologies) on 1.5% agarose gel electrophoresis.

### Yeast modeling

All yeast strains used were derived from GA74-1A (*MAT**a** his3-11,15 leu2 ura3 trp1 ade8* [*rho+ mit+*]). The *psd1Δpsd2Δ* (*MAT**a** leu2 ura3 ade8 psd1Δ::TRP1 psd2Δ::HISMX6*) was described previously (Onguka et al, 2015). The *psd1Δpsd2Δ* strain expressing WT Psd1p with a 3XFLAG tag on its C terminus enabled separate detection of the $\alpha$ and $\beta$ subunits as described (Onguka et al, 2015). The R358Q pathogenic yeast mutant was generated by overlap extension (Ho et al, 1989)

using pRS305Psd3XFLAG (Onguka et al, 2015) as the template. To generate a psd1Δpsd2Δ strain expressing the R358Q mutant, psd1Δpsd2Δ yeast were transformed with the linearized pRS305-based plasmid and genomic integrants selected on synthetic dropout media (0.17% yeast nitrogen base, 0.5% ammonium sulfate, 0.2% dropout mixture synthetic–leu, and 2% dextrose). To determine PSD1 transcript abundance, total RNA was extracted and reverse transcription–quantitative real-time PCR performed exactly as described in (Ogunbona et al, 2018) with the exception that yeast strains were cultured in YPD media at 30$^{\circ}$C and collected at an OD$_{600}$ of 0.5. ACT1 was used as a housekeeping gene. The primers used are PSD1 forward, 5′-CCAGTAGCACAAGGCGAAGA-3′; PSD1 reverse, 5′-GACAT-CAAGGGGTGGGAGTG-3′; ACT1 forward, 5′-GTATGTGTAAAGCCGGTTTTG-3′; and ACT1 reverse, 5′-CATGATACCTTGGTGTCTTGG-3′.

To determine if the R385Q mutant was functional, overnight cultures grown in YPD medium were spotted on synthetic complete dextrose plates in the absence or presence of 2 mM ethanolamine and grown at the indicated temperature. Preparation of yeast cell extracts and immunoblotting was performed as described (Claypool et al, 2006). Antibodies against the Psd1p β subunit and Pic1p were generated in the Claypool laboratory and described previously (Whited et al, 2013; Onguka et al, 2015). Other antibodies used were mouse anti-FLAG (clone M2; Sigma-Aldrich) and DyLight conjugated fluorescent secondary antibodies (Thermo Fischer Scientific).

## Supplementary Information

## Acknowledgements

The authors would like to thank the study participants and their family. We would also like to acknowledge the contributions of Drs. Ross McLeod, Rebecca Trussell, Graham Boag, Colleen Adams, Carolyn Skov, James Scott and Sheila Unger in the clinical care and previous phenotypic character-izations of this family, as well as Ms. Mary Anderson for clinical support. This work was supported by Alberta Children's Hospital Foundation (TE Shutt), the National Institutes of Health (R01GM111548 to SM Claypool), and the National Science Foundation Graduate Research Fellowship (DGE1746891 to PN Sam). This work was performed under the Care4Rare Canada Consortium funded by Genome Canada, the Canadian Institutes of Health Research, the Ontario Genomics Institute, Ontario Research Fund, Genome Alberta, Genome BC, Genome Quebec, and Children's Hospital of Eastern Ontario Foundation.

### Author Contributions

T Zhao: conceptualization, formal analysis, investigation, method-ology, and writing—review and editing.
CM Goedhart: conceptualization, data curation, formal analysis, investigation, and writing—review and editing.
PN Sam: formal analysis, investigation, and methodology.
R Sabouny: formal analysis, investigation, methodology, and writing—review and editing.
S Lingrell: formal analysis, investigation, and methodology.
AJ Cornish: resources.
RE Lamont: data curation, formal analysis, and investigation.
FP Bernier: data curation, formal analysis, and investigation.
D Sinasac: resources, data curation, and investigation.
JS Parboosingh: data curation, formal analysis, and investigation.
JE Vance: conceptualization, formal analysis, supervision, investi-gation, and writing—review and editing.
SM Claypool: conceptualization, formal analysis, supervision, in-vestigation, and writing—original draft, review, and editing.
AM Innes: conceptualization, formal analysis, supervision, investigation, project administration, and writing—original draft, review, and editing.
TE Shutt: conceptualization, formal analysis, supervision, investigation, project administration, and writing—original draft, review, and editing.

## Conflict of Interest Statement

The authors declare that they have no conflicts of interest.

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
