## [Reviewer comments · Life Science Alliance]

Life Science Alliance

PISD is a mitochondrial disease gene causing skeletal dysplasia, cataracts and white matter changes

Tian Zhao, Caitlin Goedhart, Pingdewinde Sam, Rasha Sabouny, Susanne Lingrell, Adam Cornish, Ryan Lamont, Francois Bernier, David Sinasac, Jillian Parboosingh, Jean vance, Steven Claypool, Micheil Innes, and Timothy Shutt

DOI: <https://doi.org/10.26508/lsa.201900353>

Corresponding author(s): Timothy Shutt, University of Calgary and Micheil Innes, University of Calgary

Review Timeline:	Submission Date:	2019-02-20
	Editorial Decision:	2019-02-21
	Revision Received:	2019-02-25
	Accepted:	2019-02-26

Scientific Editor: Andrea Leibfried

Transaction Report:

Please note that the manuscript was previously reviewed at other journals and the reports were taken into account in inviting a revision for publication at *Life Science Alliance* prior to submission to *Life Science Alliance*.

Referee #1 Review

Remarks for Author:

The manuscript by Zhao and colleagues reports on the discovery of mutations in the Phosphatidylserine decarboxylase (PISD) gene that are causative for a novel class of disease characterized by an impaired mitochondrial protein homeostasis, thereby resembling mitochondrial chaperonopathies. The mitochondria-localized enzyme PISD catalyzes the formation of phosphatidylethanolamine (PE) by decarboxylation of phosphatidylserine (PS). In the present study, the authors conducted whole exome sequencing (WES) on DNA isolated from two affected individuals that revealed mutations in the PISD gene. These mutations were linked to decreased conversion of the phospholipid PS to PE in patient-derived cells, and concomitantly a reduction of total cellular PE levels. Also, patient-derived fibroblasts exhibited several aspects of mitochondrial dysfunction including mitochondrial fragmentation and decreased oxygen consumption rates in Seahorse assays. Replenishing the mitochondrial pool of PE through treatment with lyso-PE restored mitochondrial morphology in patient fibroblasts. The authors then went on to perform functional assays in both yeast and human cells and found that the R277Q mutation interfered with autocatalytic processing of PISD. In addition, an analysis of mitochondrial protein homeostasis revealed effects on several mitochondrial inner membrane proteins including the protease OMA1, its substrates OPA1 and PGAM5 as well as MRPL32, a subunit of the mitochondrial ribosome.

This is a very interesting study with mechanistic data and solid experimental work that identifies a novel mitochondrial disease gene. The manuscript is well-written and features a variety of experimental systems and well-established methods. There are a few points listed below that should to be addressed in a revised version.

Major points:

1. Figure 2: Include statistical analyses in both figure panels (e.g t-test).
2. Figures 3A, C: Indicate fluorescent stain (anti-TOMM20) within the figure panels.
3. Figure 4A: Indicate fluorescent stain (anti-LAMP1) within the figure panel.
4. Figure 5: Provide control gene amplification as a loading control (e.g. RPL13A, or similar)
5. Figure 8A: The OPA1 immunoblot provided does not properly resolve the different OPA1 isoforms, and it would be interesting to see the levels of both long and short isoforms in patient cells. A 10% SDS-PAGE with low molecular weight proteins run out of the gel could help to improve the resolution.
6. Figures 7 and 8: Molecular weight markers should be added to the immunoblot images.

Minor points:

1. The manuscript lacks page numbers.
2. Update the reference format to the journal standards (e.g Authors last name et al., Year).
3. Introduction: Provide reference after first sentence ("...and numerous metabolic pathways.").
4. Introduction: Correct to ",highlighting the importance of mitochondrial PE,...

Referee #2 Review

Remarks for Author:

The authors describe PISD, encoding the phosphatidylserine decarboxylase in the mitochondrial inner membrane, as a novel mitochondrial disease gene in humans. They identified two sisters harboring compound heterozygous variants of PISD, leading to congenital cataracts, short stature and white matter changes. The pathogenic nature of the PISD mutations was verified in yeast, demonstrating that the maternal mutation affects splicing while the paternal variant impairs autocatalytic self-processing. Experiments in fibroblast show reduced PE synthesis, impaired respiration and mitochondrial fragmentation, which is consistent with previous findings after PISD knockdown. Importantly, the authors describe severe growth impairment with non-specific skeletal anomalies and dysmorphic features which are not observed in classic mitochondrial diseases. This indicates that PISD belongs to a group of genes mutated in mitochondrial chaperonopathies, which share similar symptoms. This notion is indeed intriguing and represents an important contribution to the field of mitochondrial diseases. However, the hypothesis of the authors that this is caused by an impaired mitochondrial proteostasis remains highly speculative and is the weakest part of the manuscript (Fig. 8). On one hand, mutations in most mitochondrial proteases are associated with neurodegeneration but not with skeletal abnormalities (except LON), on the other hand, mutations in mitochondrial gene expression disturb mitochondrial proteostasis and cause disease but the clinical presentation is distinct from PISD.

Specific points:

1. The authors describe different levels of mitochondrial proteases and known proteolytic substrates in PISD patient fibroblast but the molecular basis of these observations remains unclear (Fig. 8). Do mutations in PISD affect the expression of any of the analyzed genes? The authors correlate reduced OMA1 levels with impaired processing of PGAM5. However, PGAM5 is predominantly processed by PARL, which is not considered. The authors should at least use more careful wording when describing these observations.
2. The authors attribute the altered lysosomal structures in patient fibroblasts to mitochondrial deficiencies but do not explore this further. Does inactivation of PISD affect the lipid composition of lysosomal membranes?
3. Does lyso-PE suppress other mitochondrial deficiencies besides mitochondrial fragmentation?
4. Do mutations in PISD specifically affect complex IV activity as has been observed after siRNA mediated depletion of PISD?
5. The authors mention MRI scans demonstrating hypomyelination of the corpus callosum, however do not provide any data.
6. Some of the experiments lack a statistical evaluation or at least information whether or not data shown represent averages of several replicates (for instance Figs. 2, 8).
7. Fig. 3 C & D are mislabeled in the main text; Fig. 6A, 7A, 7B are seemingly not mentioned in the main text.
8. In Fig. 6 B the unprocessed form of Psd1p is not visible from the blots. How can the authors be sure that the mutant protein is expressed?

Referee #3 Review

Remarks for Author:

Zhao and colleagues describe a single family (sibling pair) subjected to whole exome sequencing in which they have identified recessively inherited (compound heterozygous) variants in the PSD gene, encoding phosphatidylserine decarboxylase. This is reported to be a mitochondrially-localised protein which functions to convert phosphatidylserine to phosphatidylethanolamine in the inner mitochondrial membrane. The authors provide some evidence for pathogenicity of the variants, show cells from one patient demonstrate evidence of mitochondrial dysfunction (Seahorse microscale oxygraphy) and present some data relating to a disturbed mitochondrial network. Addition of lyso-PE - through some mechanism - is said to rescue a network phenotype. One variant causes a splice defect, the other (missense) change is investigated in yeast and data presented in support of its role as a temperature-sensitive PSD1 allele.

There are some interesting data within the manuscript, which on the whole is well-written describing the clinical manifestations of the two affected siblings, but the functional evaluation and execution of some of the experiments to prove causality and explore a mitochondrial dysfunction require further work to a higher standard demanded of such a journal. The analysis of the patient cell line (which patient has a skin biopsy, I can't see that we are told?) is incredibly limited and there is scope for significant further work to be able to describe this as a novel mitochondrial disease gene.

Along these lines, I have some thoughts and comments for the authors:

Introduction: The Introduction to the paper generalises a number of key points about the diagnosis and growing genetic signature of mitochondrial disease - diagnosing these conditions has been tricky owing to the contribution of both the mitochondrial genome and nuclear genome but if there was a paradigm for using unbiased WES approaches coupled to post-genomic functional studies to determine mechanism and disease causality, then surely mitochondrial diseases fit the bill! The availability of high-throughput NGS has revolutionised the diagnosis of mitochondrial disease with many, many tens of novel disease genes and their associated proteins characterised in relation to a role in post-translational mitochondrial gene expression in the last 7-8 years. With regards the PSD protein, I didn't find all the necessary information I was looking for in the papers cited and wondered if in mammalian cells a mitochondrial localisations and specific mitochondrial compartment had been experimentally confirmed - I was expecting to see these experimental data in the paper (immunolocalisation and sub-mitochondrial fractionation experiments). Why does loss of PSD lead to a disturbance of the mitochondrial network - is this also seen in some of the other disorders cited (LONP1-related mitochondrial disease)?

Clinical data: What is the ethnicity of the family studied? Both individuals seem to have presented very early in life but the details of how the disease has progressed in adult life are sketchy (ie between 18-28 years in Individual 1). What were the metabolic tests performed - can more information be provided? When was a skin biopsy taken and from which sibling - not in the Methods either? When was the decision taken to subject both sisters to WES, somewhat surprising this wasn't done a number of years ago?

Variant data: be helpful to see in the Supplemental an output from the pipeline in terms of how the variants were filtered to reach the candidates tested. Have the variants been uploaded to ClinVar?

Functional characterisation of patient cells: see above - please state which sibling biopsied? It would have been helpful to look at both? Were attempts made using GeneMatcher or similar matchmaking tools to find additional families?

Additional comments and questions:

1. Figure 2B shows the Seahorse output examining basal and maximal respiration rates - a common issue throughout the manuscript is that details of numbers of experiments are not provided nor for statistical evaluations. How many times were these experiments repeated and what stats were done - we are shown error bars.

2. What is the inference of an "elevated" basal rate of respiration?

3. Why were no further experiments performed on the cells to look at other markers of mitochondrial function, especially based on what is cited in the Introduction in terms of OXPHOS defects associated with experimentally-induced loss of PISD? There are no western blotting data looking at steady-state levels of OXPHOS proteins or BN-PAGE to assess assembly of OXPHOS components (standard experiments). What about determining steady-state levels of PISD protein specifically?

4. Did the authors consider looking at mtDNA copy number given the reported effects on mitochondrial dynamics and that a loss of mtDNA copy number has been reported in association with mutation of similar proteins (mentioned in the Introduction) - see the recent work on LONP1 (<https://www.ncbi.nlm.nih.gov/pubmed/29518248>).

5. The experimental analysis and quantitation of the mitochondrial and lysosomal morphology defects are intriguing and I don't see from the methods exactly how these experiments were done; it appears that there was some sort of manual evaluation rather than using specific software to analyse the confocal images - what determines a fragmented, intermediate or fused network and how are these counted specifically? The representative images are small and could be shown at higher magnification. Is a t-test the best way to analyse the data? The authors seem to have used antibody (IF) analysis rather than conventional TMRM staining (uptake into cells dependent on the mitochondrial membrane potential)

6. The rationale for the 2DG experiment is clear but most researchers use growth in galactose; what about a simple growth curve experiment to see if cells don't grow when forced to rely on OXPHOS?

7. how does lyso-PE work? How is it preferentially sequestered and taken up into specific organelles? Was any functional assessment (Seahorse) tried post lyso-PE treatment?

8. The most compelling piece of evidence in support of pathogenicity of the PISD variants - functional complementation of the biochemical defect following retro- or lenti-viral transduction of WT PISD is not presented - even more important when only data on a single family. I think this is an absolute prerequisite to be able to publish and confirm causality of a new gene defect within a high impact research output.

Functional analyses of variants: cDNA studies - there is no annotation of the sequencing chromatograms at all in Suppl Figure 1. The yeast studies are nice, but why not look at the effect more in the human cells - do these mutations (LoF and one missense) lead to a

decreased stability of the PISD protein?

Effect on proteins involved in mitochondrial protein homeostasis: the data shown in Figure 8 are not of great quality, again how many times were these repeated and were the data reproducible? As standard, most people looking at OPA1 by western are able to identify at least 4 isoforms - why are these not present in the blots? The evidence that lyso-PE rescues the expression of these critical proteins is sketchy at best - much more important to have shown functional complementation of a full range of mitochondrial biochemical phenotypes.

Referee #2 Review

Remarks for Author:

The authors have improved the manuscript demonstrating functional complementation of PISD patient fibroblasts by wild-type PISD, confirming that the observed deficiencies are caused by the loss of PISD activity. Moreover, they have added data to demonstrate that the pathogenic mutation C266Y impairs autocatalytic processing of PISD. The manuscript therefore provides strong support for the pathogenicity of the identified PISD mutations causing skeletal abnormalities. I should add that in my opinion this analysis also extends considerably a recent report on a different heterozygous variant of PISD by Girish et al.

However, the authors did not address experimentally my main criticism on the original manuscript concerning the proposed link of PISD to the activity of mitochondrial proteases and mitochondrial chaperonopathies, which is also highlighted in the abstract. The experiments shown in Figure 9 do not support sufficiently the far-reaching claim of the authors that PISD mutations affect the activity of various mitochondrial proteases. The authors did not analyze the stability of substrate proteins nor examined proteases involved. Overexpression of wild-type and mutant PISD appears to have similar effects, an observation that remains unexplained. The quality of the analysis in Figure 9 is limited, a quantification of protein levels is missing. I agree with the authors that mitochondrial diseases show variable presentations and this holds also true for diseases associated with mitochondrial proteases. However, due to the many phenotypes, the fact that some of them overlap with those caused by PISD mutations does not provide strong support for similarities in the disease mechanisms. In my opinion, the otherwise interesting manuscript would be strengthened if the authors would not include data shown in Figure 9.

Referee #3 Review

Remarks for Author:

I previously presented a number of concerns about the data presented, and some key omissions of pertinent data.

some of these are now included (ie mitochondrial localisation data) or amended, but overall I still struggle that the authors have not undertaken a thoroughly and sufficiently deep enough analysis of the effects of PISD variation on mitochondrial function to publish in a high-impact journal; the description of another case in the literature also lessens this impact.

The complementation data - transient transfection with only 5% efficiency of gene delivery & expression - is not fully conclusive, at least based on the phenotype assessed. The authors have also suggested that fibroblasts do not express the protein at very high levels in defence of their not showing western blot data, which therefore throws into question whether the "functional" data presented are really that supportive if there is unlikely to be a strong phenotype in this cell type.

February 21, 2019

RE: Life Science Alliance Manuscript #LSA-2019-00353-T

Dr. Timothy E Shutt
University of Calgary
Medical Genetics
HMRB 268
3330 University Dr. NW
Calgary T2N 4N1
Canada

Dear Dr. Shutt,

Thank you for transferring your revised manuscript entitled "PISD is a mitochondrial disease gene causing skeletal dysplasia, cataracts and white matter changes" to Life Science Alliance. Your manuscript was previously reviewed twice at another journal, and the editors transferred those reports to us with your permission.

The reviewers thought that the revised version does not sufficiently address their request for further reaching insight into the link between PISD and the activity of mitochondrial proteases and mitochondrial chaperonopathies. This is not a concern for publication here, and we would thus be happy to publish your work in Life Science Alliance, pending a minor revision to tone down your conclusions (reviewer #2, comment regarding figure 9) and to match our formatting guidelines:

- please add callouts in the text to figure 3C, 3D, 4A, 4B, 5A-C
- all corresponding authors should link their ORCID iD to their profile within our submission system, please
- please note that we have only supplementary figures at Life Science Alliance, these will however be shown in-line in the HTML version of the paper. It would be great if you could change the callouts from EV to S figures.

A. FINAL FILES:

B. MANUSCRIPT ORGANIZATION AND FORMATTING:

Sincerely,

Andrea Leibfried, PhD
Executive Editor
Life Science Alliance
Meyerhofstr. 1

69117 Heidelberg, Germany
t +49 6221 8891 502
e a.leibfried@life-science-alliance.org
www.life-science-alliance.org

We would like to thank the reviewers for their comments and positive feedback on the relevance of our study showing that PISD is a novel mitochondrial disease gene. Their comments following the first round of revisions are included below, with our responses highlighted in red.

Referee #2 (Comments on Novelty/Model System for Author):

1. the low technical quality refers to my criticism on Figure 9

Unfortunately, we are limited by the antibodies that we have available to us. Nonetheless, in our hands, we reproducibly see the same trends in the changes in protein expression level in several mitochondrial proteins that are mediated by inner mitochondrial membrane proteins. These changes are most evident in OMA1 and MRPL32, where we see dramatic decreases in protein expression. Thus, we are confident of the results we are reporting.

2. medium novelty as a pathogenic mutation in PISD has recently been described. However, I feel that the manuscript significantly extends this study and will be of broad interest to the biomedical community.

The preprint of our article, submitted to BioRxiv, predates the manuscript by Girisha et al., which notably cites our manuscript and strengthens our findings. We also thank the reviewer for recognizing that our study will be of broad interest to the field.

Referee #2 (Remarks for Author):

The authors have improved the manuscript demonstrating functional complementation of PISD patient fibroblasts by wild-type PISD, confirming that the observed deficiencies are caused by the loss of PISD activity. Moreover, they have added data to demonstrate that the pathogenic mutation C266Y impairs autocatalytic processing of PISD. The manuscript therefore provides strong support for the pathogenicity of the identified PISD mutations causing skeletal abnormalities. I should add that in my opinion this analysis also extends considerably a recent report on a different heterozygous variant of PISD by Girish et al.

We appreciate the reviewer recognizing the value of our study, and the significant advances presented in our manuscript.

However, the authors did not address experimentally my main criticism on the original manuscript concerning the proposed link of PISD to the activity of mitochondrial proteases and mitochondrial chaperonopathies, which is also highlighted in the abstract. The experiments shown in Figure 9 do not support sufficiently the far-reaching claim of the authors that PISD mutations affect the activity of various mitochondrial proteases. The authors did not analyze the stability of substrate proteins nor examined proteases involved. Overexpression of wild-type and mutant PISD appears to have similar effects, an observation that remains unexplained. The quality of the analysis in Figure 9 is limited, a quantification of protein levels is missing.

In the first round of revision, reviewer #2 only asked for an examination of the mRNA expression for the protease targets where we see changes in expression. We included this requested data in the revised manuscript (Fig S3), which clearly demonstrate that the changes in protein expression are not due to alterations in mRNA levels. While we agree that there is still more work to be done to fully understand

the how impaired PE levels affect the various IMM proteases, we feel this is beyond the scope of the current study as it would encompass a large undertaking. This is because, due to a lack of mechanistic understanding of the many mitochondrial IMM proteases and their overlapping functions, it would not be a straightforward experiment to determine which proteases are affecting which specific substrates. Thus, we feel the experiments proposed by reviewer 2 in the second revision are too broad in scope for the current manuscript.

I agree with the authors that mitochondrial diseases show variable presentations and this holds also true for diseases associated with mitochondrial proteases. However, due to the many phenotypes, the fact that some of them overlap with those caused by PISD mutations does not provide strong support for similarities in the disease mechanisms. In my opinion, the otherwise interesting manuscript would be strengthened if the authors would not include data shown in Figure 9.

In the revised version of the manuscript we have toned down our interpretation regarding impaired protein homeostasis, and instead speculate as to a link to impaired protease activity. However, we have decided to keep Figure 9 in the revised manuscript, as we feel strongly that this novel link between PE and IMM proteases, even if not fully developed, will be of value to researchers studying mitochondrial lipids and mitochondrial proteases.

Referee #3 (Comments on Novelty/Model System for Author):

The authors have not fully responded to my earlier comments and criticism, and the presentation of transient transfection data with only 5% efficiency of transfection highlights my concerns that there is not sufficient proof beyond reasonable doubt that this is the causal gene; a lenti- or retro-viral delivery system would have been better, whilst they do not use appropriate tools to show complementation or phenotypic rescue.

The efficiency of transfection is not relevant to the rescue that we observe, as we can clearly identify which cells are transfected by immunofluorescence and use only the transfected cells for our analysis. A higher transfection efficiency would not change the results. In addition, we also show rescue with lyso-PE, as independent way to restore levels of mitochondrial PE. Moreover, the original request to perform the genetic complementation was because there were no additional reports of patients with mutations in PISD. However, with the recent report by Girisha et al., describing additional patients with mutations in PISD, there is now little doubt that PISD is responsible for the patient phenotypes.

There is also an inherent problem which now becomes evident is that the expression of this protein is likely to be very low in the cell line being studied.

Due to the low sensitivity of the PISD antibody we are using, which is the only antibody we know of that detects PISD, it is not possible to detect endogenous levels of PISD. This finding does not mean that expression of PISD is low in these cells, just that the expression is below the ability of the antibody to detect.

Referee #3 (Remarks for Author):

I previously presented a number of concerns about the data presented, and some key omissions of pertinent data. Some of these are now included (ie mitochondrial localisation data) or amended, but overall I still struggle that the authors have not undertaken a thoroughly and sufficiently deep enough analysis of the effects of PISD variation on mitochondrial function to publish in a high-impact journal; **In the revised version of the manuscript, we added several assays requested by reviewer #3 (mtDNA copy number, membrane potential, expression of OxPhos complexes) as well as additional functional assays (mitochondrial mass, complex IV activity). These new assays are in addition to mitochondrial respiration and morphological studies. Thus, we feel that we have performed a very comprehensive analysis of mitochondrial function in these fibroblast cells.**

the description of another case in the literature also lessens this impact.

We would actually argue the opposite, that additional patients with mutations in PISD actually strengthens the argument that PISD is indeed a mitochondrial disease gene.

The complementation data - transient transfection with only 5% efficiency of gene delivery & expression - is not fully conclusive, at least based on the phenotype assessed. The authors have also suggested that fibroblasts do not express the protein at very high levels in defence of their not showing western blot data, which therefore throws into question whether the "functional" data presented are really that supportive if there is unlikely to be a strong phenotype in this cell type.

These issues were addressed above.

February 26, 2019

RE: Life Science Alliance Manuscript #LSA-2019-00353-TR

Dr. Timothy E Shutt
University of Calgary
Medical Genetics
HMRB 268
3330 University Dr. NW
Calgary T2N 4N1
Canada

Dear Dr. Shutt,

Thank you for submitting your Research Article entitled "PISD is a mitochondrial disease gene causing skeletal dysplasia, cataracts and white matter changes". It is a pleasure to let you know that your manuscript is now accepted for publication in Life Science Alliance. Congratulations on this interesting work.

*****IMPORTANT:** If you will be unreachable at any time, please provide us with the email address of an alternate author. Failure to respond to routine queries may lead to unavoidable delays in publication.*******

DISTRIBUTION OF MATERIALS:

Again, congratulations on a very nice paper. I hope you found the review process to be constructive and are pleased with how the manuscript was handled editorially. We look forward to future exciting

submissions from your lab.

Sincerely,
